# Eumelanin Detection in Melanized Focal Changes but Not in Red Focal Changes on Atlantic Salmon (*Salmo salar*) Fillets

**DOI:** 10.3390/ijms242316797

**Published:** 2023-11-27

**Authors:** Kazumasa Wakamatsu, Johannes M. Dijkstra, Turid Mørkøre, Shosuke Ito

**Affiliations:** 1Institute for Melanin Chemistry, Fujita Health University, Toyoake 470-1192, Japan; sito@fujita-hu.ac.jp; 2Center for Medical Science, Fujita Health University, Toyoake 470-1192, Japan; dijkstra@fujita-hu.ac.jp; 3Department of Animal and Aquaculture Sciences, Norwegian University of Life Sciences, NO 1432 Ås, Norway; turid.morkore@nmbu.no

**Keywords:** Atlantic salmon, melanized focal changes, red focal changes, eumelanin, pheomelanin, PTCA, 4-AHP, AHPO, melanomacrophages, red blood cells

## Abstract

Superficial discolored spots on Atlantic salmon (*Salmo salar*) fillets are a serious quality problem for commercial seafood farming. Previous reports have proposed that the black spots (called melanized focal changes (MFCs)) may be melanin, but no convincing evidence has been reported. In this study, we performed chemical characterization of MFCs and of red pigment (called red focal changes (RFCs)) from salmon fillets using alkaline hydrogen peroxide oxidation and hydroiodic acid hydrolysis. This revealed that the MFCs contain 3,4-dihydroxyphenylalanine (DOPA)-derived eumelanin, whereas the RFCs contain only trace amounts of eumelanin. Therefore, it is probable that the black color of the MFCs can be explained by the presence of eumelanin from accumulated melanomacrophages. For the red pigment, we could not find a significant signature of either eumelanin or pheomelanin; the red color is probably predominantly hemorrhagic in nature. However, we found that the level of pigmentation in RFCs increased together with some melanogenic metabolites. Comparison with a “mimicking experiment”, in which a mixture of a salmon homogenate + DOPA was oxidized with tyrosinase, suggested that the RFCs include conjugations of DOPAquinone and/or DOPAchrome with salmon muscle tissue proteins. In short, the results suggest that melanogenic metabolites in MFCs and RFCs derive from different chemical pathways, which would agree with the two different colorations deriving from distinct cellular origins, namely melanomacrophages and red blood cells, respectively.

## 1. Introduction

Atlantic salmon (*Salmo salar*) is a major commercial species in the salmonid family, produced in large numbers in Norway, Chile, the UK, and Canada. Norway’s long shoreline and cold, clean seawater provide excellent conditions for its aquaculture. However, superficial hyperpigmented areas on salmon fillets, called “black spots” (BSs), represent a major problem for the commercial aquaculture of Atlantic salmon. BSs in white skeletal muscle of farmed Atlantic salmon can appear initially as red spots (RSs) characterized by hemorrhages and acute inflammation and is said to progress into BSs characterized by chronic inflammation and accumulating pigmented immune cells called “melanomacrophages” [1]. RSs and BSs are also known as “red focal changes” (RFCs) and “melanized focal changes” (MFCs) in the fillet, respectively [2] (Figure 1).

These discolorations cause large economic losses since fillets with pigmented abnormalities are commercially downgraded because of reduced attractiveness. A study found that in the production period of 59 weeks after the transfer of juvenile salmon (smolts) to sea, MFCs first appeared with low frequency but then became more frequent, whereas RFCs maintained a constant low prevalence of about 4% [3]. Over time, the prevalence of MFCs on fillets of Norwegian sea-reared salmon has increased substantially, from 7% in 2003 to 20% in 2015 [4]. Hence, the seafood farming industry is paying great attention to this problem and is seeking to understand the underlying causes of the dark discolorations on salmon fillets.

MFCs occur mainly in three locations: internal organs, the peritoneum, and the muscle fillet. Most MFCs are up to 3 cm in width and a few mm thick and are located in the front part of the fillet (Figure 1). Interestingly, although MFCs are common in farmed harvest-sized Atlantic salmon all along the Norwegian coast [5], there have not been reports that such spots are common in wild fish [1].

Melanomacrophage centers in teleost fish are distinctive groupings of phagocytic, melanin pigment-containing, immune cells that are generally found inside the stroma of tissues with immune functions; however, these centers can also develop in association with inflammation elsewhere in the fish body [6,7]. Melanomacrophages are particularly concentrated in teleost fish spleens, kidneys, and livers and can also be found in the spleens and livers of amphibians and reptiles [8,9,10,11,12,13,14,15]. In salmon, in some cases of granulomatous inflammation of the muscle, areas with pigmented melanomacrophages penetrate the peritoneum from the muscles, leading to externally visible pigmentation [5]. In the case of Piscine orthoreovirus 1 (PRV-1) infection, the appearance of melanomacrophages was coincident with—and provided a likely explanation for—the transformation from the late phase of RFCs into MFCs [1,2,3,16]. However, melanomacrophage accumulations have also been found in cases of chronic inflammation initiated by trauma without specific causative agents [17]. It has been proposed that melanomacrophages may contain several pigments, such as melanin, lipofuscin, and hemosiderin [7,18]; the presence of melanin pigment in salmon melanomacrophages was suggested through Fontana–Masson staining [19]. Gallone et al. [20] described that the melanin produced in the liver-pigmented macrophages of the frog *Rana esculenta* L. was a 5,6-dihydroxyindole (DHI)-rich eumelanin (EM), similar to *Sepia* melanin. However, to the best of our knowledge, there have not been reports regarding a biochemical and chemical analysis of melanin in melanomacrophages in fish. Gene expression studies, including the finding of tyrosinase expression in a cell line believed to represent melanomacrophages, support that melanin synthesis is the source of the dark pigment in salmon melanomacrophages and MFCs [5,17,21,22]. However, while MFCs in fillets have been associated with accumulations of melanomacrophages, they have not yet been examined using detailed chemical methods to find whether the focal changes involve melanin and the nature of that melanin.

Melanin pigments, composed of black to dark brown EM and yellow to reddish-brown pheomelanin (PM), are widely distributed in vertebrates [23]. While much is now understood about the nature of the melanin synthesized in higher vertebrates, little is known about the melanin produced by aquatic animals, including aquacultured fishes. Red seabream (*Pagrus major*), one of the most commercially valuable fish species in Japan, is usually found at around 20 m depth of seawater, and their skin is bright scarlet. However, their skin turns dark when they are bred in net cages—fish farmers call these seabreams “suntanned”—and this darkening severely reduces their market value. Adachi et al. [24] demonstrated that the cause of this “suntanning” is the production of melanin, and they were the first to chemically quantify melanin in fish. They reported that the skin of “suntanned” seabream accumulated five times higher levels of EM than the corresponding parts of shaded and wild red seabream, while PM was below detection limits in these fish [24].

Both EM and PM are derived from a common precursor, 3,4-dihydroxyphenylalanine-quinone (DOPAquinone), which is produced from tyrosine by the action of the melanogenic enzyme tyrosinase [25,26,27,28]. In the absence of thiol compounds, DOPAquinone undergoes an intramolecular cyclization of its amino group [29] to produce DOPAchrome, which is then spontaneously and gradually converted to DHI or 5,6-dihydroxyindole-2-carboxylic acid (DHICA) by tyrosinase-related protein 2 [30,31,32] or by copper ions [33]. DHI and DHICA are then further oxidized and polymerized to produce EM. Sulfhydryl compounds such as cysteine, if present at sufficient levels (>0.13 µM), may enter a route different from the normal EM pathways to give thiol adducts of DOPA, that is, 5-*S*-cysteinyldopa (5SCD) along with a minor isomer 2-*S*-cysteinyldopa [26]. Further oxidation of these thiol adducts leads to the formation of benzothiazine and benzothiazole intermediates, which then are converted to PM. In vertebrates, melanin synthesis, called “mixed melanogenesis” because it produces both EM and PM, is biochemically controlled by tyrosinase activity and cysteine concentration. The casing model of mixed melanogenesis implies that PM is always produced first, after which EM is deposited on the preformed PM [27]. In the early 1980s, to characterize melanins and melanogenesis, we developed a microanalytical method to analyze EM and PM [34] based on the chemical degradation of melanin pigments followed by analysis of the degradation products using high-performance liquid chromatography (HPLC). Later, we established a novel, more convenient method for the simultaneous measurement of EM and PM using alkaline hydrogen peroxide oxidation (AHPO) and hydroiodic acid (HI) hydrolysis [28,35,36] (Figure 2).

The AHPO generates the specific markers pyrrole-2,3,5-tricarboxylic acid (PTCA), pyrrole-2,3-dicarboxylic acid (PDCA), thiazole-2,4,5-tricarboxylic acid (TTCA), and thiazole-4,5-dicarboxylic acid (TDCA). PTCA is a specific biomarker of DHICA units or 2-substituted DHI units in EM, whereas PDCA is a specific biomarker for DHI-derived units in EM, while TTCA and TDCA are specific biomarkers for benzothiazole-derived moieties in PM. Analysis of benzothiazine-derived moieties in PM is performed using HI hydrolysis to yield 4-amino-3-hydroxyphenylalanine (4-AHP) and its isomer 3-amino-4-hydroxyphenylalanine (3-AHP) [28,34,35].

As mentioned above, previous reports have suggested that the black pigments of MFCs in salmon fillets may be melanin [2,6,19], but this assumption has been lacking chemical analysis other than histological staining [19], which, amongst others, cannot distinguish between eumelanin and pheomelanin. Therefore, for the present study, we performed the chemical analysis of MFCs and RFCs in salmon fillets using the chemical degradation methods for melanin pigments to help understand the character of MFCs and RFCs. The results revealed that the MFCs were derived from EM while the RFCs contained little or no EM or PM. The results of analyzing RFCs suggested that they included melanogenic metabolites derived from oxidized proteins produced by DOPAquinone and/or DOPAchrome binding to salmon proteins. We found support for this hypothesis by performing tyrosinase oxidation of DOPA in the presence of salmon fillet proteins. 

In short, the present study provides novel insights into the biochemistry and origin of MFCs and RFCs, which form a serious quality problem in the production of Atlantic salmon. 

## 2. Results

We used the AHPO and HI hydrolysis assays that we previously developed to investigate whether the pigments in MFCs and RFCs in salmon are produced through the pathways of melanogenesis, which begins with tyrosinase oxidation of tyrosine, and determined the melanogenic components of these pigments.

### 2.1. Chemical Characterization Using AHPO and HI Hydrolysis of MFCs and RFCs on Salmon Fillets

Salmon fillet samples prepared as described for Materials #1 and #2 from The Norwegian Institute of Food, Fisheries and Aquaculture Research (Nofima), Norway, were received in March 2020, and were analyzed. The pigmented areas had various degrees of pigmentation and were divided into three groups (small, medium, and large) depending on the size of the pigmented area: controls, MFCs (BS-small, BS-medium, and BS-large), and RFCs (RS-small, RS-medium, and RS-large) (Figure 3). Materials #1 are fillets from salmon cultivated in open cages in seawater (The LetSea Research Station, Dønna, Norway), and three pooled samples of each type of spot were analyzed. Materials #2 are fillets from salmon raised in large commercial cages hanging in sea (Lerøy Midt AS, Gjemnes, Norway), with various degrees of pigmentation (2 controls, 3 BS-small, 17 BS-medium, 7 BS-large, 1 RS-small, and 2 RS-medium).

HPLC chromatograms of the AHPO mixtures obtained for BS and RS in Materials #1 are shown in Figure 4a,b, respectively. The PTCA (a specific biomarker for DHICA-derived units in EM) values observed for MFCs were much higher than observed for RFCs, where they were barely detectable. PDCA (a specific biomarker for DHI-derived units in EM) was only detected at trace levels in both MFCs and RFCs, while TTCA and TDCA, markers of benzothiazole-derived moieties in PM, were not detected. These AHPO analysis results conclude that MFCs contain DHICA-rich EM and that RFCs contain only a trace amount of EM. HPLC chromatograms of HI hydrolysates of both MFCs and RFCs in Materials #1 show the PM markers 4- and 3-AHP which are markers of benzothiazine-derived moieties in PM, and DOPA (Figure 4c,d). The 4-AHP value was larger in RFCs than in MFCs. Although 4-AHP was detected in RFC, the PM content in RFCs was much smaller than in typical pheomelanic samples [28].

The VIS absorption spectrum of MFCs in Soluene-350 of Materials #2 was similar in pattern to that of the control solution but with increased absorbance throughout the VIS region (Figure 5a). On the other hand, the spectrum of RFCs in Soluene-350 had a large and increasing absorption below 550 nm with an absorbance five-fold greater than the control at 400 nm. The difference spectra of MFC and RFC solutions subtracted from the control spectrum are shown in Figure 5b. These difference spectra show a small and flat absorption between 400 and 650 nm in MFCs and a large and increasing absorption below 550 nm in RFCs.

The results of spectrophotometric and chemical degradation analyses of MFCs and RFCs in Material #1 fillets are summarized in Figure 6. A500 values (total melanin values) obtained by spectrophotometry, after dissolving the samples in Soluene-350, progressively increased from the control (0.0065) to BS-small (0.0073), BS-medium (0.0097), and BS-large (0.0147). The A650/A500 ratio is an indicator of whether melanin is eumelanic or pheomelanic [37]. The A650/A500 ratio of 0.22 in BS-large suggests that this melanin is eumelanic, given that there is a background absorption from the control. The increasing trend was more obvious for PTCA values with 0.7 in the control, 3.4 in BS-small, 7.0 in BS-medium, and 19.4 ng/mg in BS-large (Appendix A). This clearly supports that the black color of MFC is attributable to DOPA-derived EM. PDCA values were only at trace levels. The low PDCA/PTCA ratio in the pigment of MFC indicates that this pigment is derived from DOPA but not from DA, because DA melanin should give comparable yields of PDCA and PTCA [38]. 4-AHP and 3-AHP were detected only at trace levels in MFC, irrespective of size, indicating that MFC contain little or no PM.

The identification of PTCA in MFCs was confirmed by isolating some PTCA in a preparative scale AHPO experiment. The VIS and MS spectra are shown in Appendix A, confirming the identification of PTCA.

RFCs afforded a quite different pattern of degradation products compared with MFC (Figure 6). A500 values progressively increased from the control (0.0065), to RS-small (0.0148), RS-medium (0.0196), and RS-large (0.0208). However, the A650/A500 ratio remained similar and low (0.05 to 0.07), suggesting that RFCs contain PM or another, not yet identified melanic pigment. PTCA values were only trace (<2.0 ng/mg), irrespective of size, indicating that this pigment is not a typical EM. 4-AHP values in RFCs were many-fold higher than in MFCs. Nevertheless, the low absolute values of 4-AHP (0.5–0.7 ng/mg) in RFCs indicate that PM does not constitute a major portion of RFCs. To characterize the pigment of MFCs, we also analyzed DOPA after HI hydrolysis (named HI-DOPA). HI-DOPA can be produced from free DOPA, from free cysteinyldopa (CD) isomers, or from protein-bound (PB) DOPA and CD isomers [39,40]. HI-DOPA values (2.4–4.7 ng/mg) in MFCs did not differ from the control (4.7 ng/mg). In contrast, HI-DOPA values progressively increased in RFCs from the control (4.7), to RS-small (10.3), RS-medium (15.3), and RS-large (17.0 ng/mg). A similar trend was found for PB-5-*S*-cysteinyldopa (5SCD), a metabolite of DOPA, that arises from the conjugation of DOPAquinone to sulfhydryl groups of proteins [39,40] (Figure 6). While PB-5SCD values in MFCs remained the same (2.1–2.6 ng/mg) as the control (2.3 ng/mg), those in RFCs were two-fold higher (4.3–5.2 ng/mg) than the control. The above results indicate that the degree of pigmentation of RFCs correlates well with the amount of melanin metabolites (HI-DOPA, PB-5SCD).

Analyses of Materials #1 led to the conclusion that the pigments of MFCs are eumelanic while the pigments of are RFCs are neither eumelanic nor pheomelanic. We, therefore, wished to confirm these results using a greater number of MFCs (a total of 27) from another set of materials (Materials #2) obtained from salmon raised in large cages in the sea.

Figure 7 summarizes the results of those analyses. The results show that pigments in MFC (n = 27) from Materials #2 had a similar pattern of melanin markers, e.g., A500, PTCA, 4-AHP, and HI-DOPA, as those in Materials #1. Only the high value of HI-DOPA in the control is puzzling (Figure 7, Appendix A). Taking advantage of analyzing a large number of MFCs, we could assess the correlation between A500 and PTCA values. As shown in Appendix A, the PTCA values correlated well (R^2^ = 0.754) with the A500 values with a background value of 0.006/mg, indicating that the black color is mostly due to DOPA-derived EM. The RFCs in Material #2 were only available from three fillets, but the melanin marker values in these RFCs were similar to those in Materials #1.

Regarding the chemical characterization of RFCs, it is noteworthy that the average 4-AHP/3-AHP ratio in three RFCs (Materials #1) was 0.77 (Appendix A); this suggests that a melanic pigment was produced by interactions between DOPAquinone and NH_2_ groups in salmon proteins (Figure 8). This interaction would lead to the formation of a Schiff’s base between one of the two carbonyl groups in DOPAquinone and the protein amino group [41]. Reductive acid hydrolysis of such proteins with HI would give rise to the production of nearly equal amounts of 4-AHP and 3-AHP, as the 4-AHP/3-AHP ratio should be much greater than 1 in PM [35]. Thus, we hypothesized that the pigment of RFCs includes a form of oxidized proteins produced by DOPAquinone and/or DOPAchrome binding to proteins in salmon fillets. To confirm this hypothesis, we then performed an experiment mimicking the production of RFCs in salmon fillets.

### 2.2. RFCs Are Suggested to Include Melanogenic Metabolites Derived from Oxidized Proteins Produced by Dopaquinone and/or DOPAchrome Binding to Salmon Proteins

It is known that during a 4 h oxidation of DOPA by tyrosinase, all DOPA is oxidized to DOPAquinone, which is rapidly converted to the red pigment DOPAchrome that gives rise to EM via DHI and DHICA [25,26]. A minor amount of DOPAquinone may undergo conjugation with sulfhydryl or amino groups in proteins [41]. 

To replicate this, mushroom tyrosinase was added to a mixture of a salmon fillet homogenate (20 mg/mL) + DOPA (1 mM), DOPA (1 mM) alone, and salmon fillet homogenate (20 mg/mL) alone, and the reaction mixtures were incubated at pH 7.4 and 37 °C for 4 h followed by measuring the melanin markers. The results of this mimicking experiment are summarized in Figure 9. The A500 value of 0.0128/mg after tyrosinase oxidation of the salmon + DOPA group was intermediate between the DOPA alone (DOPA-melanin) group (0.0189/mg) and the salmon alone group (0.0067/mg) (Figure 9a). The A650/A500 ratio in the salmon + DOPA group (0.110) was closer to the salmon alone group (0.050) than the DOPA alone group (0.317). These results indicate that the oxidation of DOPA to DOPA-melanin was suppressed in the salmon + DOPA group. This can be explained by interactions between salmon proteins and DOPA oxidation products. The PDCA value (37.3 ng/mg) in the salmon + DOPA group was higher than that of DOPA alone (17.5 ng/mg), suggesting that formation of DHI-protein conjugates derived from the conjugation of DOPAquinone or DOPAchrome and salmon proteins (Figure 9b). This suggests the presence of DHI-protein conjugates or related adducts in the salmon + DOPA group (Figure 8).

The 4-AHP value (1.6 ng/mg) in the salmon + DOPA group was very low, indicating that the contribution of PM can be ignored (Figure 9c). The 3-AHP value (1.7 ng/mg) in the salmon + DOPA group was also low, and the 4-AHP/3-AHP ratio of 0.9 had the same tendency as that of RFC in salmon (Figure 6). The salmon + DOPA group gave a higher yield of 4-AHP in comparison to the DOPA alone and the salmon alone groups. This suggests that 4-AHP is derived from interactions between DOPAquinone and the sulfhydryl and/or amino group in salmon proteins, since the tyrosinase oxidation of DOPA (no cysteine available) yielded no level of 4-AHP [26,41]. The DOPA value (648 ng/mg, 33% yield) following HI hydrolysis (HI-DOPA) in the salmon + DOPA group was much higher than in the DOPA alone group (12 ng/mg) or the salmon alone group (128 ng/mg) (Figure 9d). Similarly, the PB-5SCD value (685 ng/mg, 22% yield) in the salmon + DOPA group was much higher than in the DOPA alone group (0.5 ng/mg) or the salmon alone group (100 ng/mg) (Figure 9d). The high values of HI-DOPA and PB-5SCD in the salmon alone group may be ascribed to the oxidation of tyrosine residues in salmon proteins by tyrosinase. We have previously shown that tyrosine residues in proteins can be oxidized by tyrosinase to DOPA residues and then to CD residues [42]. These results indicate that a considerable amount (33%) of the DOPA oxidized (1970 ng/mg) was consumed by binding with proteins but was not oxidized to DOPA melanin. As free DOPA (unoxidized DOPA) was at a negligible level, DOPA produced by HI hydrolysis should be derived from PB- DOPA and CD [39,40]. This is consistent with the 22% yield of PB-5SCD from the salmon + DOPA group. This indicates that PB-5SCD can be considered as a marker of DOPA oxidation (to DOPAquinone). The binding of DOPA with proteins through sulfhydryl groups leads to a lower production of DOPA melanin, which is consistent with the lower A500 value in the salmon + DOPA group compared with the DOPA alone group. Thus, the above experimental results are consistent with the suggestion that RFCs include oxidized proteins produced by DOPAquinone and/or DOPAchrome binding to salmon proteins.

## 3. Discussion

The results of the present study clearly indicate that the black pigment of the MFCs is EM derived from DOPA. To the best of our knowledge, this is the first time that this has been clearly shown using detailed chemical analytical methods. This EM is probably derived from melanomacrophages which are immune cells that may have entered the muscle for a number of possible reasons. We are not aware that there has been a proper biochemical characterization of the black pigment of fish melanomacrophages yet. Although we did not investigate isolated melanomacrophages directly, our study suggests that their pigment is EM because they are believed to give the black color to MFCs.

In contrast, the origin of pigment of the RFCs was less obvious. The levels of 4-AHP, HI-DOPA, and PB-5SCD in RFCs were higher than in MFCs and were proportional to the degree of coloration (Figure 6). Based on these results, the red pigment appeared to include conjugations of DOPAquinone and/or DOPAchrome with salmon fillet proteins. To prove this hypothesis, we performed an experiment mimicking the process of RFCs production. The exposure of salmon fillet proteins to DOPAquinone (or to DOPAchrome) resulted in the suppression of melanin production and the production of pigment producing PDCA, 4-AHP, HI-DOPA, and PB-5SCD. Thus, we observed the characteristic features of RFCs in the mimicking experiment except for the PDCA production, which remains puzzling. The production of 4-AHP (and 3-AHP) may be explained by Schiff’s base formation between DOPAquinone and the amino group in lysine residues in salmon proteins [41]. PM was detected at only trace levels in MFCs, irrespective of size, indicating that MFCs contain little or no PM. This may be because salmon muscle tissue has little cysteine levels compared to other amino acids [43,44], so the availability of cysteine in muscle tissue is limited. Since cysteine is required for the production of pheomelanin, low cysteine levels in salmon muscle are thought to lead to decreased PM production. This is consistent with the previous results that PM in red seabream was below the detection limits [24].

The absolute values of HI-DOPA and PB-5SCD were 40–100 times higher in the salmon + DOPA group than in RFCs (Figure 9d). These differences may be ascribed to the fact that DOPA is rapidly oxidized by tyrosinase in the mimicking experiment, whereas in RFCs in salmon, the oxidation (production of pigment of RFCs) only gradually progresses over a long period of time. The level of PDCA in the salmon + DOPA group in the mimicking experiment was two-fold higher than from the DOPA alone and the salmon alone groups combined (Figure 9b). This PDCA may arise from DHI-protein conjugates formed via cyclization of PB-CD. The reason why PTCA and PDCA values in RFC are similar to those of the control is unknown, but this may also be due to secondary changes in DHI-protein conjugates giving a reddish coloration of RFC (Figure 8). 

The spectrum of RFCs in Soluene-350 showed a significant increase in absorption below 550 nm, and the absorbance at 400 nm was five-fold greater than the control at 400 nm (Figure 5). As Soluene-350 has a large absorption below 400 nm, the maximum of absorption spectrum could not be measured, but the large absorption of RFCs at 400 nm indicates the existence of quinone with an absorbance maximum at 400 nm or DHI-protein conjugates with nearly 400 nm.

Knowing that MFC pigment consists of melanin may help to reduce MFC numbers by reducing melanin synthesis. For example, from studies in mammals it is known that unsaturated fatty acids decrease melanin synthesis and tyrosinase activity whereas saturated fatty acids increase those activities [45,46]. In a recent study, salmons fed a diet with increased n-3 long-chain polyunsaturated fatty acids, eicosapentaenoic acid, and docosahexaenoic acid, had a lower occurrence of black spots [47,48,49].

The etiology of the focal melanization in the white muscle is complex and probably can include a variety of reasons that induce tissue damage (which can cause intramuscular bleeding) and/or immune responses (which can cause melanomacrophages to infiltrate the muscle). The aquaculture industry has also suggested a correlation between RFCs (also known as “bleedings”) and MFCs, and transient forms have been observed [3]. Our experimental results, however, do not support the possibility that melanin pigment from MFCs is a final product of the pigment found in RFCs; our experimental result agrees with the common idea that RFCs and MFCs are derived from different cell types, namely erythrocytes and melanomacrophages. Our current hypothesis is that, in regard to the biochemistry of their melanogenic metabolites, MFCs and RFCs branch into two chemical pathways in a common biosynthetic pathway of melanogenesis that leads to the production of EM in MFCs, and melanogenic metabolites derived from PB-DOPAquinone and/or DOPAchrome in RFCs (shown schematically in Figure 8).

Lastly, we will consider the involvement and role of ascorbic acid and red blood cell (RBC) oxidation on living organisms. To reduce lipid oxidation, researchers have explored adding antioxidants, like vitamin E, to fish diets or injecting mixtures of antioxidants (ascorbic acid, citric acid, and selenium) into fish fillets. Interestingly, the reaction of ascorbic acid and protein has been reported to produce a red color under aerobic conditions, which was shown to result from an amino–carbonyl reaction of oxidized ascorbic acid (dehydroascorbic acid) [50]. The iron in hemoglobin may contribute to changes in pigment intensity in RFCs. It is known that RBC are exposed to reactive oxygen and nitrogen species, which can lead to the production of oxidants [51]. Oxyhemoglobin in RBCs can undergo autoxidation to form active oxygen (superoxide, hydrogen peroxide, and so on). Free hemoglobin is particularly toxic, and this is evident in several RBC diseases [52]. This oxidation process can also cause the oxidation of tyrosine residues, leading to the formation of dityrosine, dopamine, dopamine quinone, DHI, and other related products [53]. When considering the production of RFCs, it is important to note that further research and experiments may be necessary to fully understand and confirm these reactions and their implications.

## 4. Materials and Methods

### 4.1. Fish Material

The fish examined were clinically healthy farmed Atlantic salmon from two different populations. From both populations (Material #1 and #2), fillets were selected from the processing line, specifically focusing on focal discolored spots located in the cranio-ventral part of the fillets. Stained tissue was carefully excised, and adjacent unstained muscle tissue located 3 cm posterior to the discolored area was also cut out and utilized as a control. Samples within the same category were combined, frozen at −80 °C and shipped to Fujita Health University in Japan in March 2020, for chemical characterization. Our control samples were sampled as normal muscle from another location without pigmented areas after removing the pigmented areas from the fillet of salmon with pigmented areas. Hyperpigmented muscle (RFCs and MFCs) was sampled and trimmed for unstained tissue, while control muscle was sampled 3 cm beside the pigmented spots.

Materials #1: Atlantic salmon with known pedigree, were obtained from the breeding nucleus of Benchmark Genetics, Norway (average body weight 4.2 kg). Out of the 2005 registered fish at Austevoll Laksepakkeri, Norway, 32% had focal dark spots on at least one of their fillets. In total, dark-stained tissue samples were collected from 636 fillets.

The pigmented areas had various degrees of pigmentation. According to the color, black and red spots were separated. Pigmented spots with the same macroscopic appearance were divided into three groups (small, medium, and large) depending on the pigment intensity and pooled: controls, MFCs (BS-small, BS-medium, and BS-large) and RFCs (RS-small, RS-medium, and RS-large) (Figure 6). Three pellets of pigmented areas from each group of MFCs and RFCs were dried in a desiccator (dry weights, 90 to 160 mg) and were homogenized at a concentration of 20 mg/mL in water using a Ten-Broeck glass homogenizer. Aliquots of 100 µL (2 mg) were subjected to AHPO to measure PTCA and PDCA [36], to HI hydrolysis to measure 4-AHP, 3-AHP, and HI-DOPA [35,39] and to solubilization in Soluene-350 to measure A500 and A650 [37]. PB-5SCD was analyzed after precipitation of proteins with 0.4 M HClO_4_ followed by HCl hydrolysis, as described in [40]. 

Materials #2: Atlantic salmon (average weight 4 kg) farmed in open commercial-sized sea cages (Lerøy Midt AS, Gjemnes, Norway) [54]. The fish were filleted at the Lerøy processing plant on Hitra island, Norway. Salmon fillets, dissected from melanized or RFC (31 spotted samples and 2 controls; 250 to 350 mg) were dried in a desiccator (dry weights, 90 to 160 mg) and homogenized at a concentration of 20 mg dry weight/mL in water using a Ten-Broeck glass homogenizer. Aliquots of 100 µL (2 mg) were analyzed as described for Materials #1.

Mushroom tyrosinase (1715 U/mg) and L-DOPA were purchased from Sigma-Aldrich (St Louis, MO, USA). Soluene-350 was purchased from PerkinElmer (Waltham, MA, USA). Other chemicals are of the highest purity commercially available.

### 4.2. HPLC Conditions for the Determination of Specific Markers, PTCA, PDCA, 4-AHP, 3-AHP, HI-DOPA, and PB-5SCD

A HPLC system consisting of an analytical UV/VIS detector, a JASCO pump (JASCO Co., Tokyo, Japan), a C18 column (Capcell Pak MG; 4.6 × 250 mm; 5 µm particle size, Osaka Soda, Osaka, Japan) and a JASCO UV-visible detector (JASCO Co., Tokyo, Japan) was used to measure PTCA, and PDCA [36]. The mobile phase was 1 mM tetra-*n*-butylammonium bromide in 0.1 M potassium phosphate buffer (pH 2.1): methanol, 83:17 (*v*/*v*) [55]. Analyses were performed at 40 °C at a flow rate of 0.7 mL/min. UV-visible spectra were measured using a JASCO V-630 UV-VIS spectrophotometer (JASCO Co., Tokyo, Japan). For the assay of 4-AHP, 3-AHP, and DOPA (HI-DOPA) in the HI hydrolysate, a Catecholpak C18 column was used with 4-AHP buffer—methanol, 98:2 (*v*/*v*) at 35 °C, with an electrochemical detector set at +500 mV versus an Ag/AgCl electrode. The AHP buffer consisted of 0.1 M sodium citrate buffer, pH 3.0, containing 1 mM sodium octanesulfonate and 2% EDTA.2Na [35]. PB-5SCD was analyzed as described for serum 5SCD [40]. High-resolution MS spectra were obtained using a 6220 TOF mass spectrometer (mode: electrospray ionization-time-of-flight, negative; ESI (-)-TOF) (Agilent Technologies, Santa Clara, CA, USA). Units of Specific markers, PTCA, PDCA, 4-AHP, 3-AHP, HI-DOPA, and PB-5SCD, were expressed as ng/mg.

### 4.3. Preparative Isolation of PTCA from MFC

Preparative scale AHPO was performed. MFCs from Materials #2 (wet weight 2.0 g) were combined and homogenized in 50 mL 1 M K_2_CO_3_, to which 10 mL 30% H_2_O_2_ was added. After stirring for 20 h at 25 °C, 1.5 g Na_2_SO_3_ and 23 mL 6 M HCl were added to stop the oxidation and acidify the mixture to pH < 2. After removal of proteins by filtration, the oxidation mixture was extracted 3 times with 100 mL ethyl acetate. The ethyl acetate was removed in vacuo and the residue was dissolved in 0.4 M HCOOH:MeOH, 90:10, and injected into a C18 column (Capcell Pak MG; 20 × 250 mm; 5 µm particle size, Osaka Soda, Osaka, Japan). HPLC separation was performed with a mobile phase of 0.4 M HCOOH:MeOH, 90:10 at a column temperature of 45 °C and a flow rate of 7 mL/min. Fractions containing PTCA were recovered in a total yield of 18 µg. A PTCA standard (30 µg) was also injected into the HPLC and the fraction containing PTCA was recovered for comparison of UV spectra and high-resolution MS spectra. High-resolution MS 198.0036 [M-H]^−^, calculated for C_7_H_4_O_6_N_1_, [M-H]^−^, 198.1044.

### 4.4. Experiment Mimicking the Production of RFC in Salmon Fillets

Solutions containing: (1) 0.2 mM L-DOPA and 20 mg salmon fillet homogenate; (2) 0.2 mM L-DOPA alone; or (3) 20 mg salmon fillet homogenate alone in 1 mL 50 mM sodium phosphate buffer, pH 7.4, were oxidized by mushroom tyrosinase (50 U) at 37 °C. After vigorous mixing for 4 h, aliquots of each oxidation mixture were analyzed for melanin markers. Aliquots of 200 µL were mixed with 800 µL Soluene-350 and were then analyzed for A500 and A650 [37]. Aliquots of 100 µL were subjected to AHPO to analyze PDCA and PTCA, to HI hydrolysis to measure 4-AHP, 3-AHP, and HI-DOPA [35], and to HClO_4_ precipitation followed by HCl hydrolysis to measure PB-5SCD [39,40]. Melanin marker values were normalized per mg salmon protein. 

### 4.5. Statistical Analysis

Student’s *t*-test (two tailed) was performed using Microsoft Excel for Mac (Ver 16.79.1 (23111614), Japan Microsoft Co., Tokyo, Japan). A *p*-value of <0.05 was considered statistically significant.

## 5. Conclusions

Comprehending the essential processes linked to hyperpigmented focal spots in salmon fillets is crucial as a basis for implementing measures to minimize the prevalence of this expensive quality issue. This study is the first to have elucidated the melanogenic features of the pigments in MFCs and RFCs by detailed chemical analytical methods. The analytical results indicate that the pigment of MFCs is DOPA-derived EM, and that the pigment of RFCs, which is pigment produced by intramuscular hemorrhage, includes melanogenic metabolites derived from the oxidized proteins produced by DOPAquinone and/or DOPAchrome binding to salmon proteins. From our results, it can be deduced that the pigment in melanomacrophages (the source of MFCs) is DOPA-derived EM, which was anticipated but nevertheless fills an important gap in the knowledge of the immune system of primitive vertebrates. Although RFCs have been reported to change into MFCs [1,3], our data show that the pathways of melanogenesis in the production of MFCs and RFCs are distinct, agreeing with a different biochemical and cellular origin of MFCs and RFCs pigments. When we consider any hypotheses explaining why focal melanization occurs specifically in cultured fishes in cages, muscle injury or rupture are a plausible initial trigger for focal melanization. However, the underlying reason for the higher deposition of melanin pigment in injured tissue of salmon raised in large open sea cages compared with small research units, and the absence of this phenomenon being common in wild salmon and salmon farmed in freshwater [54], remains unknown. Possibly, higher abundances of inflammatory mediators, e.g., due to the higher presence of bacteria and/or viruses in large open sea cages, promote melanogenesis. In order to reduce RFCs and MFCs, future research should try to learn how to prohibit hemorrhages and (other) inflammations in the muscle tissue of farmed salmon, possibly by addressing injuries from handling, high population density, nets, and/or reducing the pathogen load from eutrophicated cage water.

## Figures and Tables

**Figure 1 ijms-24-16797-f001:**
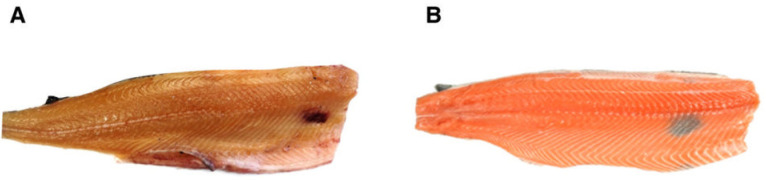
The typical position of (**A**) red spots (red focal changes (RFCs)) and (**B**) black spots (melanized focal changes (MFCs)) on Atlantic salmon fillets.

**Figure 2 ijms-24-16797-f002:**
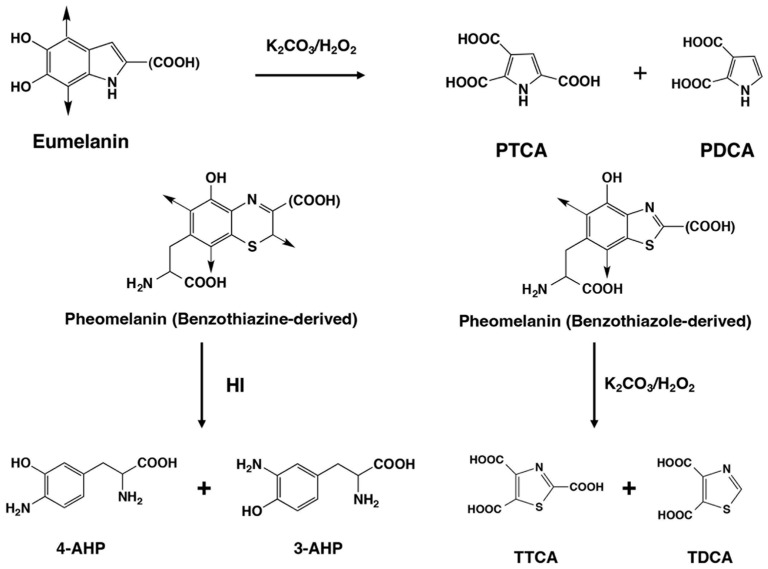
Summary of chemical degradation by AHPO and HI hydrolysis of EM and PM. EM consists of DHI and DHICA (with a carboxyl group) units. For details, see the introduction.

**Figure 3 ijms-24-16797-f003:**
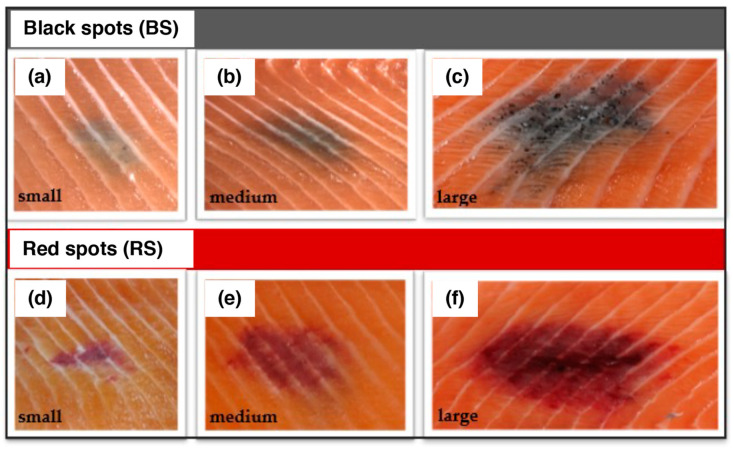
Macroscopic grading of focal black spots (BS) (**a**–**c**) and red spots (RS) (**d**–**f**) of the cranioventral section of Atlantic salmon fillets (Material #1). The spots were classified into three categories based on their size (diameter) and pigmentation: small/weak stained spots (<3 cm), medium-sized spots (3 cm) with clear discoloration, and large spots (3–6 cm) with distinct discoloration.

**Figure 4 ijms-24-16797-f004:**
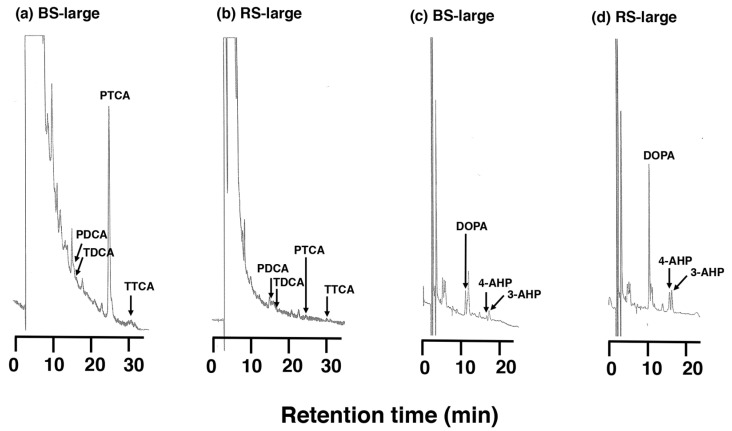
HPLC chromatograms of AHPO (**a**,**b**) and HI hydrolysis (**c**,**d**) of BS-large (**a**,**c**) and RS-large (**b**,**d**), respectively.

**Figure 5 ijms-24-16797-f005:**
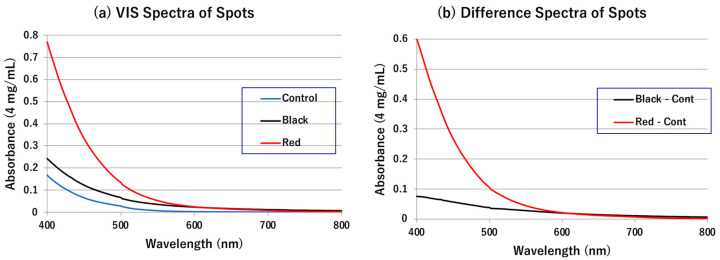
(**a**) UV–vis absorption spectra of MFC and RFC solutions in Soluene-350. (**b**) Difference spectra of MFC and RFC solutions that are subtracted from the control spectrum, respectively.

**Figure 6 ijms-24-16797-f006:**
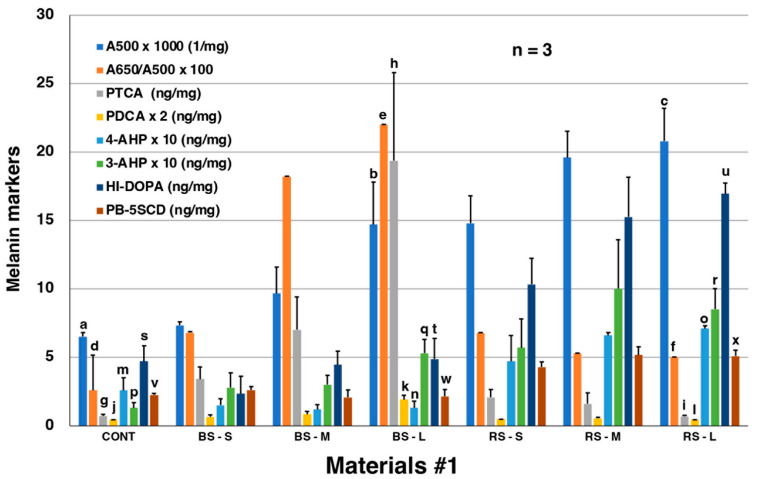
Summary of spectrophotometric and chemical degradation analyses of control, MFC (BS-small, BS-medium, and BS-large) and RFCs (RS-small, RS-medium, and RS-large) in Materials #1. The number of samples in each group = 3. Since A500, A650/A500 ratios, PDCA, 4-AHP, and 3-AHP were small values, these values were multiplied by 1000, 100, 2, 10, and 10, respectively. Error bars represent one standard error from the mean. Statistical analysis was performed by the unpaired Student’s *t*-test (two-tailed). Statistical significant differences: *p* < 0.001 (e–f; n–o; s–u; t–u), *p* < 0.01 (a–c; d–e; j–k; k–l; m–o; p–r; v–x), *p* < 0.05 (a–b; g–h; g–i; h–i; p–q; w–x), Non-significant difference; *p* > 0.3 (d–f; s–t; v–w), *p* > 0.1 (b–c; j–l; m–n; q–r).

**Figure 7 ijms-24-16797-f007:**
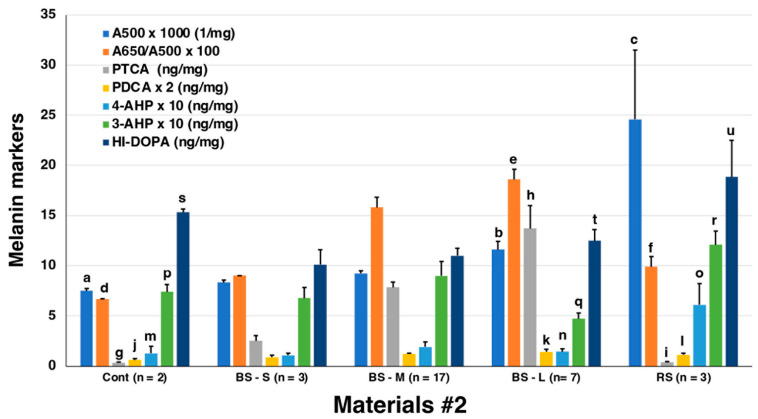
Summary of spectrophotometric and chemical degradation analyses of control (n = 2), MFCs (BS-small, n = 3; BS-medium, n = 17; BS-large, n = 7) and RFCs (1 RS-small, 2 RS-medium, n = 3) in Materials #2. As A500, A650/A500 ratios, PDCA, 4-AHP, and 3-AHP showed small values, these values were multiplied by 1000, 100, 2, 10, and 10, respectively. Error bars represent one standard error from the mean. Statistical significant differences: *p* < 0.0005 (q–r), *p* < 0.01 (d–e; e–f; h–i; n–o), *p* < 0.05 (a–b; d–f; g–h; p–g; t–u), non-significant differences; *p* < 0.1 (p–r; q–i), *p* > 0.1 (a–c; b–c; j–k; j–l; m–o; s–t), *p* > 0.3 (k–l; m–n; s–u).

**Figure 8 ijms-24-16797-f008:**
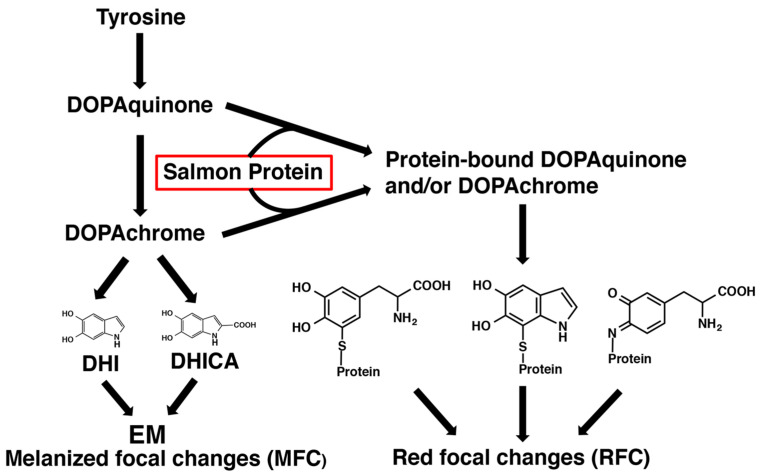
Our current hypothesis of the biosynthetic pathway of the melanogenic components of MFCs and RFCs. The oxidation of DHI and DHICA via DOPAchrome forms MFCs (EM). On the other hand, the PB-DOPAquinone and DOPAchrome formed by the reaction of DOPAquinone and DOPAchrome with salmon proteins produce PB-DOPA and PB-DHI included in RFCs.

**Figure 9 ijms-24-16797-f009:**
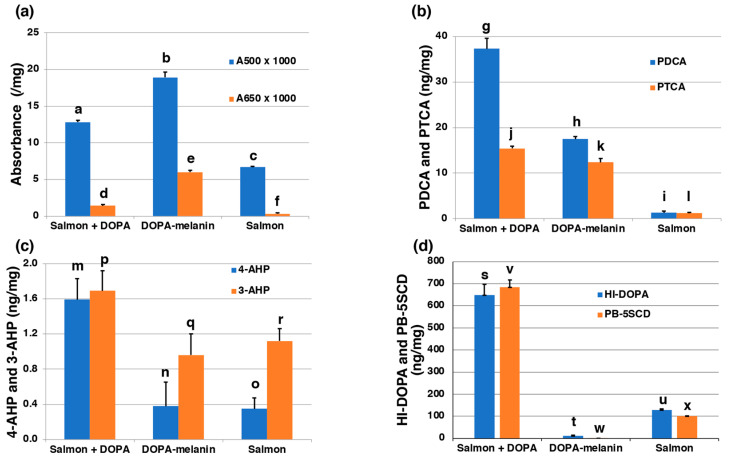
Summary of the mimicking experiment using Salmon + DOPA, DOPA-melanin, and Salmon. (**a**–**d**) are the absorbance value, PDCA and PTCA values, 4-AHP and 3-AHP values, and HI-DOPA and PB-5SCD values, respectively. Mushroom tyrosinase was added to mixtures of salmon fillet homogenate (20 mg/mL) + DOPA (1 mM), DOPA (1 mM) alone, and salmon fillet (20 mg/mL) alone followed by measuring the melanin markers. Error bars represent one standard error from the mean. Experiments were performed in triplicate. Statistical significant differences: *p* < 0.0001 (a–c; b–c; d–e; d–f; g–i; h–i; j–l; k–l; s–t; s–u; t–u; v–w; v–x; w–x), *p* < 0.0005 (a–b; g–h; m–o, *p* < 0.005 (j–k; m–n), *p* < 0.05 (e–f; p–q; p–r). Non-significant difference; *p* < 0.3 (n–o; q–r).

## Data Availability

Data is contained within the article and Appendix A.

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
