# Peer review of "Eumelanin Detection in Melanized Focal Changes but Not in Red Focal Changes on Atlantic Salmon (Salmo salar) Fillets"

_ijms, 2023, doi:10.3390/ijms242316797_

Round 1

Reviewer 1 Report

Comments and Suggestions for Authors

I think that for a better understanding, the methodology part should be included after the introduction.

The conclusions should be extended much more although they clearly leave out results.

The experiments should be supported with a statistical study that is not described.

Comments on the Quality of English Language

is ok

Author Response

<Reviewer 1>

We highly appreciate your time and efforts to help us improve our article on the biochemistry of black and red spots in salmon filets. Your kind advice has led to major improvements.

Apart from answering your comments, in the new version, the previous Figures 6, 7, and 9 have been newly drawn.

Please find below our responses to the individual comments of the Reviewers.

Comments and Suggestions for Authors

  • I think that for a better understanding, the methodology part should be included after the introduction.

We understand the preference of the Reviewer. However, the paper format in IJMS specifies that chapters should be presented in the order: 1. Introduction, 2. Results, 3. Discussion, 4. Materials and Methods, and 5. Conclusion. Therefore, we would like to maintain the chapters in their current order.

  • The conclusions should be extended much more although they clearly leave out results.

We thank the Reviewer for this critical comment. Although we like to keep the Conclusions section succinct, we have now extended the contents to improve the readability. The chapter now says:

“5. Conclusions

Comprehending the essential processes linked to hyperpigmented focal spots in salmon fillets is crucial as a basis for implementing measures to minimize the prevalence of this expensive quality issue. This study is the first to have elucidated the melanogenic features of the pigments in MFC and RFC by detailed chemical analytical methods. The analytical results indicate that the pigment of MFC is DOPA-derived EM, and that the pigment of RFC, which are pigments produced by intramuscular hemorrhage, includes melanogenic metabolites derived from the oxidized proteins produced by DOPAquinone and/or DOPAchrome binding to salmon proteins. From our results, it can be deduced that the pigment in melanomacrophages (the source of MFC) is DOPA-derived EM, which was anticipated but nevertheless fills an important gap in the knowledge of the immune system of primitive vertebrates. Although RFC have been reported to change into MFC [1,3], our data show that the pathways of melanogenesis in the production of MFC and RFC are distinct, agreeing with a different biochemical and cellular origin of MFC and RFC pigments. When we consider any hypotheses explaining why focal melanization occurs specifically in cultured fishes in cages, muscle injury or rupture are a plausible initial trigger for focal melanization. However, the underlying reason for the higher deposition of melanin pigment in injured tissue of salmon raised in large open sea-cages compared with small research units, and the absence of this phenomenon being common in wild salmon and salmon farmed in freshwater [53], remains unknown. Possibly, higher abundances of inflammatory mediators, e.g. due to the higher presence of bacteria and/or viruses in large open sea-cages, promote melanogenesis. In order to reduce RFC and MFC, future research should try to learn how to prohibit hemorrhages and (other) inflammations in the muscle tissue of farmed salmon, possibly by addressing injuries from handling, high population density, and nets, and/or reducing the pathogen load from eutrophicated cage water.”

  • The experiments should be supported with a statistical study that is not described.

We thank the Reviewer for pointing this out. We have now added statistical analysis results to Figures 6, 7, and 9. In Figure 6, we investigated the statistical analysis between Control, BS-L, and RS-L. On the other hand, in Figure 7, statistical analysis was performed between Control, BS-L, and RS. These results were shown in the figure legend of each figure.

Furthermore, we have now added the following sentence to Materials and Methods:

“4.5. Statistical analysis

Student’s t-test (two-tailed) was performed using Microsoft Excel for Mac (Japan Microsoft Co., Tokyo, Japan). A p-value of < 0.05 was considered statistically significant.”

Comments on the Quality of English Language is ok.

Reviewer 2 Report

Comments and Suggestions for Authors The authors performed chemical characterization of melanized focal changes, and also of red pigment (called red focal changes (RFC)), of salmon fillets using alkaline hydrogen peroxide oxidation and hydroiodic acid hydrolysis. The results highlighted different chemical pathways for melanogenic metabolites in MFC and RFC. The whole work can be quite meaningful for quality control of salmon fillet products. Generally, this is an interesting topic, and the results, figures are sufficient. It can be published with a few modifications:

1) The significant difference should be specified during data analysis

2) The abstract should be summarized, and also can stand alone, without confusion.

3) The conclusion cannot be simply a overview of the results, and further research as well as the application on industry can also be added.

  Comments on the Quality of English Language

The authors performed chemical characterization of melanized focal changes, and also of red pigment (called red focal changes (RFC)), of salmon fillets using alkaline hydrogen peroxide oxidation and hydroiodic acid hydrolysis. The results highlighted different chemical pathways for melanogenic metabolites in MFC and RFC. The whole work can be quite meaningful for quality control of salmon fillet products. Generally, this is an interesting topic, and the results, figures are sufficient. It can be published with a few modifications:

1) The significant difference should be specified during data analysis

2) The abstract should be summarized, and also can stand alone, without confusion.

3) The conclusion cannot be simply a overview of the results, and further research as well as the application on industry can also be added.

Author Response

<Reviewer 2>

We highly appreciate your time and efforts to help us improve our article on the biochemistry of black and red spots in salmon filets. Your kind advice has led to major improvements.

Apart from answering your comments, in the new version, the previous Figures 6, 7, and 9 have been newly drawn.

Please find below our responses to the individual comments of the Reviewers.

Comments and Suggestions for Authors

 The authors performed chemical characterization of melanized focal changes, and also of red pigment (called red focal changes (RFC)), of salmon fillets using alkaline hydrogen peroxide oxidation and hydroiodic acid hydrolysis. The results highlighted different chemical pathways for melanogenic metabolites in MFC and RFC. The whole work can be quite meaningful for quality control of salmon fillet products. Generally, this is an interesting topic, and the results, figures are sufficient. It can be published with a few modifications:

 We thank the Reviewer for the kind words regarding our results being sufficient and having practical relevance.

1) The significant difference should be specified during data analysis

We thank the Reviewer for pointing this out. We have now added statistical analysis results to Figures 6, 7, and 9. In Figure 6, we investigated the statistical analysis between Control, BS-L, and RS-L. On the other hand, in Figure 7, statistical analysis was performed between Control, BS-L, and RS. These results were shown in the figure legend of each figure.

Furthermore, we have now added the following sentence to Materials and Methods:

“4.5. Statistical analysis

Student’s t-test (two-tailed) was performed using Microsoft Excel for Mac (Japan Microsoft Co., Tokyo, Japan). A p-value of < 0.05 was considered statistically significant.”

2) The abstract should be summarized, and also can stand alone, without confusion.

We point out that the Abstract has only 221 words, which is a quite normal length. Furthermore, although we understand that this is a personal perception, we feel that the abstract is easy to read and that all the important elements of the study are explained. Also because the other two reviewers did not comment on this, we prefer to keep the Abstract as it is.

3) The conclusion cannot be simply a overview of the results, and further research as well as the application on industry can also be added.

We thank the Reviewer for this critical comment. Although we like to keep the Conclusions section succinct, we have now extended the contents to improve the readability and relevance. The chapter now also discusses how MFC and RFC research should continue, and says:

“5. Conclusions

Comprehending the essential processes linked to hyperpigmented focal spots in salmon fillets is crucial as a basis for implementing measures to minimize the prevalence of this expensive quality issue. This study is the first to have elucidated the melanogenic features of the pigments in MFC and RFC by detailed chemical analytical methods. The analytical results indicate that the pigment of MFC is DOPA-derived EM, and that the pigment of RFC, which are pigments produced by intramuscular hemorrhage, includes melanogenic metabolites derived from the oxidized proteins produced by DOPAquinone and/or DOPAchrome binding to salmon proteins. From our results, it can be deduced that the pigment in melanomacrophages (the source of MFC) is DOPA-derived EM, which was anticipated but nevertheless fills an important gap in the knowledge of the immune system of primitive vertebrates. Although RFC have been reported to change into MFC [1,3], our data show that the pathways of melanogenesis in the production of MFC and RFC are distinct, agreeing with a different biochemical and cellular origin of MFC and RFC pigments. When we consider any hypotheses explaining why focal melanization occurs specifically in cultured fishes in cages, muscle injury or rupture are a plausible initial trigger for focal melanization. However, the underlying reason for the higher deposition of melanin pigment in injured tissue of salmon raised in large open sea-cages compared with small research units, and the absence of this phenomenon being common in wild salmon and salmon farmed in freshwater [53], remains unknown. Possibly, higher abundances of inflammatory mediators, e.g. due to the higher presence of bacteria and/or viruses in large open sea-cages, promote melanogenesis. In order to reduce RFC and MFC, future research should try to learn how to prohibit hemorrhages and (other) inflammations in the muscle tissue of farmed salmon, possibly by addressing injuries from handling, high population density, and nets, and/or reducing the pathogen load from eutrophicated cage water.”

Reviewer 3 Report

Comments and Suggestions for Authors

 Wakamatsu et al. explored the cause of discolored spots on Atlantic salmon fillets—melanized focal changes (MFC) and red focal changes (RFC). Chemical analysis reveals that MFC's black color is due to eumelanin from melanomacrophages, while RFC's red color is likely hemorrhagic, with pigmentation linked to DOPAquinone and/or DOPAchrome conjugations.

This work addresses an acute and industrially significant problem and provides valuable insights for its resolution. I recommend publishing this manuscript after addressing the following minor issues:

 (i) Consider including measurements from a normal tissue region without any discolored spots when conducting analytic methods (AHPO and HI hydrolysis). This would serve as essential control data.

 (ii) Have you considered any hypotheses explaining why focal melanization occurs specifically in cultured fishes, possibly in cages? Exploring this biological question could add interest and potentially provide a key to solving the problem at hand.

Comments on the Quality of English Language

none

Author Response

<Reviewer 3>

We highly appreciate your time and efforts to help us improve our article on the biochemistry of black and red spots in salmon filets. Your kind advice has led to major improvements.

Apart from answering your comments, in the new version, the previous Figures 6, 7, and 9 have been newly drawn.

Please find below our responses to the individual comments of the Reviewers.

Comments and Suggestions for Authors

 Wakamatsu et al. explored the cause of discolored spots on Atlantic salmon fillets—melanized focal changes (MFC) and red focal changes (RFC). Chemical analysis reveals that MFC's black color is due to eumelanin from melanomacrophages, while RFC's red color is likely hemorrhagic, with pigmentation linked to DOPAquinone and/or DOPAchrome conjugations.

This work addresses an acute and industrially significant problem and provides valuable insights for its resolution. I recommend publishing this manuscript after addressing the following minor issues:

We are very pleased with these positive comments by the Reviewer!

  • Consider including measurements from a normal tissue region without any discolored spots when conducting analytic methods (AHPO and HI hydrolysis). This would serve as essential control data.

We believe that our controls, the results of which are shown in the figures, were sufficient. As described in "Materials and Methods," our control samples were sampled as normal muscle from another location without pigmented areas after removing the pigmented areas from the fillet of salmon with pigmented areas. Hyperpigmented muscle (RFC, MFC) was sampled and trimmed for unstained tissue, while Control muscle was sampled 3 cm beside the pigmented spots.

  • Have you considered any hypotheses explaining why focal melanization occurs specifically in cultured fishes, possibly in cages? Exploring this biological question could add interest and potentially provide a key to solving the problem at hand.

We thank the Reviewer for drawing attention to this. We have now added the following sentence to the Conclusion section:

“When we consider any hypotheses explaining why focal melanization occurs specifically in cultured fishes in cages, muscle injury or rupture are a plausible initial trigger for focal melanization. However, the underlying reason for the higher deposition of melanin pigment in injured tissue of salmon raised in large open sea-cages compared with small research units, and the absence of this phenomenon being common in wild salmon and salmon farmed in freshwater [53], remains unknown. Possibly, higher abundances of inflammatory mediators, e.g. due to the higher presence of bacteria and/or viruses in large open sea-cages, promote melanogenesis.”

Comments on the Quality of English Language is none.